# Effects of Different Varieties on Physicochemical Properties, Browning Characteristics, and Quality Attributes of *Mume fructus* (Wumei)

**DOI:** 10.3390/foods13091377

**Published:** 2024-04-29

**Authors:** Lei Gao, Hui Zhang, Hui Wang, Tao Wang, Aichao Li, Hongmei Xiao, Yihao Liu, Zhian Zheng

**Affiliations:** 1College of Engineering, China Agricultural University, Beijing 100083, China; 2College of Food Science and Technology, Nanjing Agricultural University, Nanjing 210095, China; 3Institute of Traditional Chinese Medicine, Mianyang Hospital of Traditional Chinese Medicine, Mianyang 621053, China

**Keywords:** *Mume fructus*, drying processing, variety, browning mechanism, quality evaluation

## Abstract

The dried *Mume fructus* (MF) is a special food and herbal medicine with a long history of processing and application. The browning index (*BI*) of *Prunus mume* (PM) is pivotal in determining the medicinal value and edible quality of MF. Exploring the BI of PM holds significant importance for both the selection of PM varieties and understanding the formation mechanism of high-quality MF. This study systematically analyzed the physicochemical properties, *BI*, and quality indicators of four PM varieties (Qingzhu Mei, Yesheng Mei, Nangao Mei, and Zhaoshui Mei) after processing into MF. The results showed significant differences in eight physicochemical indicators among the four PM varieties (*p* < 0.05). Notably, Qingzhu Mei exhibited the highest titratable acid content, Nangao Mei had the most prominent soluble solid and soluble sugar content, and Zhaoshui Mei showed outstanding performance in reducing sugar, soluble protein, and free amino acids. Regarding drying characteristics, Yesheng Mei and Nangao Mei required a shorter drying time. In terms of *BI*, Nangao Mei exhibited the greatest degree of browning and its color appearance was darker. When considering quality evaluation, Nangao Mei excelled in rehydration ability and extract content, while Zhaoshui Mei demonstrated outstanding levels of total phenols, total flavonoids, and total antioxidant capacity. Overall, the evaluation suggested that the Nangao Mei variety was more suitable for MF processing. These research results provide a valuable theoretical foundation for understanding the *BI* of PM varieties and serve as a reference for the selection of PM varieties as raw materials for processing MF.

## 1. Introduction

*Mume fructus* refers to the dried, nearly mature fruit of *Prunus mume* Sieb. et Zucc. [1], which is categorized as a medicinal and food homologous product in China’s original Ministry of Health publication. In terms of medicinal value, it has multifaceted benefits, including astringent lung and cough-relieving properties, the ability to constrict the intestine and halt diarrhea, detumescence and detoxification, promoting body fluid production and alleviating thirst, and calming effects against ascarids [2,3]. Furthermore, modern pharmacological studies have revealed additional benefits of MF, including its anti-viral hepatitis, anti-tumor, antibacterial, anti-inflammatory, and anti-fatigue properties [4]. MF contains a diverse range of components, such as organic acids, terpenes, sugars, phenols, amino acids, and alkaloids, among others [5], all of which contribute to its numerous health benefits. Notably, MF exhibits a relatively high content of organic acid. The 2020 Chinese Pharmacopoeia establishes citric acid content as the quality control standard for MF [1]. The raw material used for processing MF is PM (also called plum); PM encompasses numerous cultivars, such as Nangao Mei, Qingzhu Mei, Daqing Mei, Zhaoshui Mei, Yeisheng Mei (original variety), and so on [6]. These cultivars are widely distributed in various regions worldwide, including China, Japan, Korea, and Southeast Asia.

But it can be processed into MF only in China. The primary production regions of MF are mainly located in Sichuan, Yunnan, Fujian, and Zhejiang [7].

In recent years, a lot of research has been conducted on the processing of PM, mainly including hot-air drying [7], convection drying [8,9], freeze drying [10], microwave drying [11], and smoke processing [12]. Among them, the quality evaluation of dehydrated foods has always been of special concern, especially those processed into MF. The prevailing belief is that MF exhibiting darker hues, such as black or brown, is indicative of superior quality [13]. It is found that a series of browning reactions typically occur during the processing of PM into MF [12], and the degree of the reaction is closely related to processing conditions such as temperature and relative humidity. Moreover, it is also associated with the content and structure of the internal chemical components of the raw materials [14,15,16]. However, the variety plays a crucial role in influencing the content and structure of these internal components of PM, including textural characteristics of PM fruit [4,17], types and content of organic acids [18], protein content, total phenols, total flavonoid content [6], etc. Furthermore, studies have also revealed significant variations in the chemical composition of processed MF (different varieties of PM) originating from different regions. For instance, Chen et al. found that the citric acid content of Zhejiang PM processed into MF was notably higher than that of Sichuan MF [19]. While Huang et al. analyzed aspects such as the pulp rate and organic acids of Wu Mei, they concluded that the quality of Fujian MF was better than that of Sichuan, Zhejiang, and Yunnan [20]. It can be seen that there are significant differences in the chemical and structural compositions of different PM varieties, as well as disparities in quality indicators among MF sourced from different origins. However, due to the long process of MF processing and market circulation, and many influencing factors such as origin, variety, and processing method, there is insufficient research on the impact of a single indicator of PM variety on the quality of processed MF, and there are few relevant references.

In China, numerous PM varieties are used as raw materials to be processed into MF, which are both medicinal and edible. However, there are few professional studies on which varieties are the most suitable for this purpose. It is of great significance to explore the impact of different PM varieties on the production of high-quality MF. The main objectives of this study are (1) to explore the physicochemical properties of different PM varieties and their impact on processing characteristics; (2) to examine the changes in internal effective components pre- and post-processing, and reveal the browning mechanism during processing; and (3) to evaluate the influence of different PM varieties on the quality of processed MF and identify the suitable PM varieties for MF processing.

## 2. Materials and Methods

### 2.1. Raw Materials

Four kinds of fresh PM were chosen and are depicted in Figure 1. Among them, Qingzhu Mei (QZM) was harvested from the PM plantation in Zhao’an County, Fujian Province (116°55′ E–117°22′ E, 23°35′ N–24°10′ N). Yesheng Mei (YSM) and Nangao Mei (NGM) were harvested from Weigan Village in Pingwu County, Sichuan Province (103°50′ E–104°58′ E, 31°59′ N–33°02′ N). Zhaoshui Mei (ZSM) was harvested from the PM planting area in Eryuan County, Yunnan Province (99°32′ E–100°20′ E, 25°41′ N–26°16′ N). To maintain freshness, the four types of PM were stored in a refrigerator at a temperature of 4 ± 1 °C and 90% RH in the drying laboratory of the College of Engineering, China Agricultural University. It was ensured that the selected PM samples for the experiment were undamaged, had a firm texture, and were of moderate size to ensure consistency in the test materials. There were 4 days between harvest and processing. Prior to processing, the PM was taken out of the refrigerator and allowed to equilibrate at room temperature for approximately 30 min to achieve uniformity in temperature with the surrounding environment. Subsequently, any remaining fruit stems were removed from the PM using toothpicks. The PM was then rinsed thoroughly 2–3 times with tap water. After rinsing, the PM was wiped clean with absorbent paper and placed in a ventilated area for drying. The moisture content of the fresh PM was calculated according to the AOAC method [21]. The PM surfaces were evenly cut with blades and then dried in a vacuum dryer (D27-6050, Jinghong Instrument Co., Ltd., Shanghai, China) at 70 °C for 24 h.

### 2.2. Equipment Procedure

This study utilized the temperature and humidity precise control hot-air drying (THPC-HAD) equipment (ICTHI-150, Instrument Equipment Co., Ltd., Shanghai, China) located at the College of Engineering, China Agricultural University, to process the PM. Its schematic diagram is shown in Figure 2 and consists of an internal circulation fan (turbulence fan), heating pipes, a humidification and dehumidification device, temperature and humidity sensors, a compressor, and a control system [22]. The temperature control accuracy of this equipment was ±0.1 °C, and the humidity control accuracy was ±1.5% RH.

Prior to the experiment, the dryer was turned on and operated for 15 min to reach the set process parameters. Then, 500 ± 5 g of fresh PM was evenly distributed on the tray rack of the drying chamber in one layer, which was covered with a food-grade silicone pad. The “three-stage variable process parameter” was adopted in this study. In the first stage, the processing temperature was 80 °C, with a relative humidity of 50%, aiming to achieve a moisture content of 50% in the fresh PM material. Subsequently, the temperature was maintained and the relative humidity was adjusted to 60%, and the processing time was 24 h. Finally, the temperature was reduced to 60 °C, the humidity was set to the moisture removal mode, and drying was carried out until reaching the safe moisture content specified in the 2020 edition of the Chinese Pharmacopoeia, which is 16% [1,23]. The processed MF was marked as QZM-MF, YSM-MF, NGM-MF, and ZSM-MF.

### 2.3. Moisture Content (MC)

The moisture content of PM on a wet basis was calculated by the following formula [24]:(1)$$M_{T}=\frac{W_{t}-G}{W_{t}}$$
where *M_T_* is the moisture content of PM on a wet basis in g/g; *W_t_* is the total mass of PM processing at time *t* in g; *G* is the mass of dry matter in PM in g.

### 2.4. Physicochemical Property Measurement

The fresh PM was peeled and pitted, and the obtained pulp was homogenized into a slurry using a mortar and pestle. The pH, total soluble solids (TSSs), titratable acids (TAs), soluble sugars (SSs), reducing sugars (RSs), soluble proteins (SPs), amino acids (ACs), and citric acid (CA) content of PM were measured following the method described by AOAC [21,25], and all the above tests were conducted in triplicate. The pH value was measured using a digital pH instrument; the TSS content was assessed using a handheld refractometer and expressed as a mass fraction (%). The SS content was determined by the anthrone reagent method, while the RS content was assessed by the 3,5-dinitrosalicylic acid method. The SP content was determined using the Coomassie Brilliant Blue method, and the AC content was measured by the ninhydrin colorimetric method. Additionally, the CA content was determined by a colorimetric method, while the TA content was expressed based on the CA content using Formula (2):(2)$$TA=\frac{C\times V\times K}{m}\times \frac{V_{1}}{V_{2}}\times 100(\%)$$
where *C* is the concentration of NaOH solution in mol/L; *V* is the volume of a standard NaOH solution consumed by titration in mL; *m* is the mass of the sample in g; *V*_1_ is the volume of sample solution taken during titration in mL; *V*_2_ is the total volume of the sample diluent in mL; *K* is the grams of citric acid equivalent to 1M NaOH, with K = 0.064.

### 2.5. Color Measurement

The color parameters of PM and processed MF were determined using a colorimeter (SMY-2000SF, Shengmingyang Co., Beijing, China) [26]. The *L**, *a**, and *b** represent whiteness/blackness, redness/greenness, and yellowing/blueness, respectively. Each sample was evaluated from three consistent directions, and the resulting values were averaged. That was one repetition. Each type was repeated three times. The total color difference (Δ*E)* was calculated using Equation (3):(3)$$\Delta E=\sqrt{{\left(L_{0}^{*}-L^{*}\right)}^{2}+{\left(a_{0}^{*}-a^{*}\right)}^{2}+{\left(b_{0}^{*}-b^{*}\right)}^{2}}$$

The browning index (*BI*) of PM was calculated using Equation (4):(4)$$BI=100\times \left(\frac{X-0.31}{0.17}\right)$$
where
(5)$$X=\frac{\left(a^{*}+1.75L^{*}\right)}{\left(5.645L^{*}+a^{*}-3.012b^{*}\right)}$$
where *L*_0_***, *a*_0_*, and *b*_0_*** are the color parameters of fresh PM; *L**, *a**, and *b** are the color parameters of processed MF.

### 2.6. Rehydration Ratio and Microstructure

The rehydration ratio (RR) is used to assess the water absorption capacity of processed dried MF samples. A higher RR value indicates a stronger rehydration capability of the dried product. Firstly, approximately 20.0 g of dried MF was placed into a glass beaker filled with 100 mL of distilled water. Then, the beaker was heated and rehydrated in a 100 °C water bath. At intervals of 10 min, the MF samples were weighed after wiping off surface moisture with filter paper. This process was continued until a consistent weight of the MF sample was achieved after being weighed twice consecutively. All experiments were conducted in triplicate. The RR of MF was calculated using Equation (6) [27]:(6)$$RR=\frac{M_{e}}{M_{0}}$$
where *RR* is the rehydration ratio, and *M_e_* and *M*_0_ are the weight of materials after and before rehydration, respectively, in g.

The microstructure of the MF surface was examined using a scanning electron microscope (SEM) (SU3500, Hitachi, Japan). MF samples were cut into slices measuring 4.0 mm × 4.0 mm × 2.0 mm using a new blade and then coated with a layer of gold for 30 s. Observations were carried out at an acceleration voltage of 10.0 kV, with magnifications set at 100 times, 500 times, and 2000 times, respectively. At least 9 images were collected for each MF sample.

### 2.7. Texture Measurement (TM)

According to the measurement method of Yang et al. with slight modifications [28], the MF sample was placed on a TA-XT texture analyzer (Brookfield Engineering Laboratories, Inc. Middleboro, MA, USA) for structural profile analysis. A P/35 probe was selected, and the parameters of the pre-test speed, test speed, and post-test speed were set at 5.0 mm/s, 1 mm/s, and 2 mm/s, respectively. The compression times were separated by a 10 s interval, with a sample compression deformation of 10% and a trigger point load of 20 N. Hardness was determined based on the maximum force value during the initial compression. Ten MF samples were chosen randomly for each experiment, and the results were averaged.

### 2.8. 5-Hydroxymethylfurfural (5-HMF) Measurement

Following the method outlined by Aktag et al. with slight modifications [29], the content of 5-hydroxymethylfurfural (5-HMF) in MF was determined. Initially, the dried MF was peeled off the pulp; then, the pulp was crushed into powder with a small crusher (FTT-2500T, Dongguan Fangtai Electric Co., Ltd., Dongguan, China) and passed through a 40-mesh sieve. Then, 0.5 g of MF powder was thoroughly mixed with 2 mL of ultrapure water to form the sample solution. Subsequently, 0.22 g of the sample was weighed and put into a 2 mL centrifuge tube containing 500 μL of a 20% methanol aqueous solution. The mixture was vortexed for 1 min, sonicated for 20 min, and centrifuged at 6000 rpm for 5 min at 4 °C. Next, 1 mL of ethyl acetate was added to the centrifuge tube, followed by vortexing for 1 min and centrifugation at 6000 rpm for 5 min. The ethyl acetate layer was then transferred to another centrifuge tube and then evaporated and concentrated at 40 °C until dry. Next, 100 μL of methanol and 0.1% formic acid water were added to dissolve the residue, followed by centrifugation at 4000 rpm for 2 min and filtration through a 0.22 μm membrane. Finally, a high-performance liquid chromatography system (UPLC, Waters 2695, Agilent Co., Ltd., Santa Clara, CA, USA) equipped with a Waters 2996 UV detector was employed for quantitative analysis of the 5-HMF content in MF. The analysis conditions were as follows: The chromatographic column was a Diamonsil C18 (250 × 4.5 mm, 5 μm). The mobile phase comprised 0.1% formic acid and methanol in a ratio of 80:20, with a flow rate of 0.5 mL/min and an injection volume of 5 μL. The column temperature was maintained at 35 °C, and the absorbance of the test solution was measured at 284 nm.

### 2.9. Intrinsic Quality Indicators

#### 2.9.1. Determination of Extract Content

The determination of the water-soluble extract in MF was conducted in accordance with the method outlined in the 2020 Chinese Pharmacopoeia [1]: 2.00 g of MF powder was weighed and mixed with 50 mL of deionized water. The mixture was placed in a 150 mL conical flask and allowed to stand for 1 h. Subsequently, the content of water-soluble extract in MF was calculated using the hot leaching method, considering the sample in its dry state. Each experiment was repeated three times, and the average value was recorded.

#### 2.9.2. Determination of Citric Acid (CA) Content

The determination of CA content was performed following the method described in the 2020 Chinese Pharmacopoeia, with slight modifications [1,7]. Firstly, 0.2 g of MF powder was added to a beaker containing 50 mL of deionized water. The mixture was then heated and refluxed for 1 h, followed by cooling. In cases where weight loss occurred, appropriate amounts of deionized water were added to compensate. The mixture was thoroughly shaken and filtered, passing the filtrate through a 0.22 μm aqueous phase membrane (PES membrane). Finally, CA content determination was carried out using a high-performance liquid chromatography (HPLC) system (Waters Technology Co., Ltd., Shanghai, China) equipped with a Waters 2689 UV detector. The analysis conditions were as follows: The chromatographic column was an Inertsil ODs-3, 4.6 × 250 mm (GL Sciences Inc., Shanghai, China), with the column temperature set at 30 °C. The mobile phase consisted of 0.5% ammonium dihydrogen phosphate/acetonitrile (97:3, pH adjusted to 3 with phosphoric acid), with a flow rate of 1 mL/min. The injection volume of the reference substance was 10 μL, while that of the test sample was 5 μL. The elution time was 30 min. The CA content was calculated based on the dried product.

#### 2.9.3. Determination of Total Phenol (TP) Content

The TP content of MF was determined according to the Folin phenol method as described by Zhou et al. with relevant adjustments [30]: firstly, weighing out 30 mg of MF powder, then adding 1.5 mL of 60% ethanol and extracting it by shaking at 60 °C for 2 h. Afterward, the mixture was centrifuged at 25 °C and 12,000 rpm for 10 min. The supernatant was taken and diluted to 1.5 mL with 60% ethanol, which was the MF extract. Then, 40 μL of MF extract was mixed with 200 μL of F-C reagent, and we let it stand for 3 min in a dark environment at room temperature (25 °C) to act as the sample. Subsequently, 200 μL of a Na_2_CO_3_ saturated solution and 360 μL of distilled water were added to the sample, which was left to stand for 30 min in the dark at room temperature (25 °C). Finally, the TP content was determined by measuring the absorbance at 760 nm with a UV-1800 spectrophotometer (Shimadzu, Kyoto, Japan). The TP content in MF was calculated based on the equivalent of gallic acid per gram of dry matter (mg GAE/g DM). Each experiment was conducted in triplicate.

#### 2.9.4. Determination of Total Flavonoid (TF) Content

The TF content in MF was determined using the NaNO_2_-Al(NO_3_)_3_-NaOH spectrophotometric method described by Gao et al. with slight modifications [31]. Firstly, 200 μL of the MF extract was mixed with 60 μL of NaNO_2_ reagent and allowed to stand at room temperature (25 °C) for 6 min to serve as the sample. Subsequently, 120 μL of Al(NO_3_)_3_ reagent was added to the sample, shaken well, and left at room temperature (25 °C) for 6 min. And then, 420 μL of NaOH reagent was added, shaken well, and left to stand at room temperature (25 °C) for 15 min. The absorbance was measured at 510 nm using the spectrophotometer (Shimadzu, Kyoto, Japan) to determine the TF content in the MF extract. The TF content was calculated based on the rutin equivalent per gram of dry matter (mg CAE/g DM). Each experiment was performed in triplicate.

#### 2.9.5. Total Antioxidant Capacity (TAC)

The TAC of MF was determined using the iron ion reduction/antioxidant capacity (FRAP) method as described by Benzie et al. with slight modifications [32]. Firstly, 75 μL of MF extract was mixed with 75 μL of distilled water and 850 μL of FRAP reagents. The mixture was shaken well and allowed to stand at room temperature (25 °C) for 10 min. Subsequently, the absorbance value at 590 nm was measured using a spectrophotometer (Shimadzu, Kyoto, Japan) to determine the TAC of MF. The TAC of MF was calculated based on the Trolox equivalent per gram of dry matter (mg TE/g DM). Each experimental group was replicated three times.

### 2.10. Statistical Analyses

SPSS statistical software (version 21.0, SPSS Inc., Chicago, IL, USA) was used to analyze the experimental data. The data are expressed as the mean ± standard deviation of three determinations, and the significant differences among the PM varieties processed by different methods were determined followed by one-way analysis of variance (ANOVA) with Duncan’s test (*p* < 0.05). Origin 2020 software (OriginLab Corp., Northampton, MA, USA) were used for data processing and chart drawing.

## 3. Results and Discussion

### 3.1. Analysis of Physicochemical Properties among Different PM Varieties

The physical and chemical properties of raw materials play pivotal roles in determining the quality of processed products [33]. Among them, variety and origin are the two most important indicators that affect the physical and chemical properties of raw materials. As shown in Table 1, there are significant differences in chemical composition among the four varieties of PM (*p* < 0.05). Among them, QZM exhibited the highest moisture content (85.89 ± 0.91%), possibly attributed to the suitable temperature (21.3 °C) and abundant rainfall (1700 mm) in its growing environment [34]. At the same time, the TA content of QZM (6.61 ± 0.20 g/100 g FW) was significantly higher than that of YSM, NGM, and ZSM (*p* < 0.05), resulting in its lowest pH value. But from the perspective of CA content, this may be related to its variety characteristics. NGM demonstrated notable advantages in terms of total soluble solids (10.90 ± 0.10%) and soluble sugar content (33.09 ± 0.45 mg/g FW). This is possibly due to the suitable temperature (19.2 °C) and sufficient light during its growth [35]. ZSM exhibited higher levels of reducing sugar (4.84 ± 0.06 mg/g FW), soluble protein (3.26 ± 0.12 mg/g FW), and amino acid (6.54 ± 0.11 u mol/g FW) contents, which may be attributed to the lower temperature (13.9 °C) and moderate precipitation (732 mm) during its growth period [36], as well as its inherent variety characteristics. Moreover, a comparison between YSM and NGM, both situated in the same planting area of identical origin, revealed significant differences in chemical composition (*p* < 0.05). This illustrated the pivotal played role by variety in chemical characteristics. In conclusion, both variety and origin exerted significant influences on the quality of fresh PM.

### 3.2. Analysis of Drying Characteristics among Different PM Varieties

MF processing is one of the most crucial methods of utilizing PM. Using THPC-HAD equipment, PM samples were processed into MF via the same three-stage variable process (the first stage: 80 °C, 40% RH; the second stage: 80 °C, 60% RH, 24 h; the third stage: 60 °C, moisture removal mode). The initial moisture content of the fresh PM samples is shown in Table 1. After the drying process, the moisture content of MF was about 16%. Specifically, the moisture content of QZM, YSM, NGM, and ZSM was 15.87 ± 0.43%, 15.91 ± 3.57%, 15.68 ± 1.56%, and 15.95 ± 1.54%, respectively. Curves of the moisture content and total drying time of the different varieties of PM are shown in Figure 3. The results indicate that the total drying time of different PM varieties was different. YSM had the shortest drying time, which was 43 h, while the drying times of NGM, QZM, and ZSM were 45 h, 49 h, and 57 h, respectively. Compared with YSM, the drying time increased by 4.65%, 13.95%, and 32.56%, respectively. Figure 3 shows that the drying curves of NGM, QZM, and YSM were similar. However, as depicted in Table 1, the initial moisture content of NGM and QZM was significantly higher than that of YSM, which potentially accounted for the prolonged drying time. The initial moisture content of ZSM and YSM was almost the same. Notably, the drying rate of ZSM decreased significantly in the second stage of the reaction. This is probably because of its relatively compact texture, which resulted in slower diffusion of internal water during the browning reaction stage characterized by high humidity, consequently prolonging the drying time. Wang et al. also found similar conclusions during the drying process of grapes [26]. Furthermore, it has been reported that factors such as cell structure, chemical composition, and moisture distribution can also affect the drying rate [37].

### 3.3. Rehydration Ratio and Microstructure among Different MF Varieties

Rehydration ability refers to the capacity of fresh fruits and vegetables to absorb water and regain their original freshness after drying, and it is an important indicator for assessing the quality of dried products [38]. The RR of MF samples processed from different PM varieties with THPC-HAD is shown in Figure 4. The results revealed that the RR of the four dried MF samples ranged from 56.83% to 67.63%. Significant differences in rehydration ability were observed among different MF samples (*p* < 0.05), with NGM-MF samples exhibiting the highest rehydration ability, followed by ZSM-MF samples, with QZM-MF samples showing the lowest rehydration ability. There was no significant difference in rehydration ability between YSM-MF and ZSM-MF samples (*p* > 0.05). To explore the reasons for these differences, SEM was used to examine the surface microstructure of the processed MF samples. As shown in Figure 5, SEM analysis revealed that the QZM-MF samples had a thicker peel, closely interconnected layers, a smooth surface, and minimal pore grooves, which hindered water infiltration and resulted in the lowest rehydration ability [39]. YSM-MF samples featured fewer pores and grooves, but the mesocarp layer was loose with larger gaps, which was conducive to water infiltration and enhanced its rehydration ability. ZSM-MF samples had a thick peel and a compact mesocarp layer, with numerous twisted pores and grooves in the longitudinal direction, which could enhance the water permeability and improve its rehydration ability. This was consistent with the research by Zhou et al., which concluded that dried Chinese wolfberry with more microporous structures on the surface had stronger water permeability and rehydration ability [40]. The fruit skin layer in NGM-MF samples was relatively loose, but the fruit skin was the thinnest, which was very conducive to water penetration, resulting in its strongest rehydration ability. This was consistent with the study of Zhang et al. [41].

### 3.4. Analysis of Texture Characteristics (TPA) among Different Varieties of PM and MF

Textural characteristics are important sensory indicators for evaluating the appearance quality of food, which directly affect the taste profile and stability of food [42,43]. Meanwhile, the texture of raw materials can have a direct or indirect effect on the processing characteristics and textural properties of dried products [44], mainly including hardness, elasticity, cohesion, chewiness, gumminess, and recovery [45]. The texture characteristics of fresh PM and processed MF samples from different varieties are shown in Table 2. There were significant differences in hardness, gumminess, chewiness, and resilience among fresh PM of different varieties (*p* < 0.05). ZSM had the highest hardness, gumminess, chewiness, and resilience, while QZM had the lowest hardness, chewiness, and resilience and NGM had the lowest hardness. The hardness, cohesion, gumminess, chewiness, and resilience of processed MF samples were significantly higher than those of fresh PM samples. This was mainly due to the fact that the moisture was removed during the processing of PM into MF, and the internal substances were concentrated, resulting in a tighter texture. Additionally, the intense browning reaction during processing caused certain components to polymerize and solidify, further contributing to the increased texture parameters of MF [46]. There were notable differences (*p* < 0.05) observed in the hardness, cohesion, gumminess, chewiness, and resilience of different MF samples, which were mainly caused by the different texture characteristics of processed raw materials, according to Table 2. ZSM-MF had the highest hardness, elasticity, cohesiveness, gumminess, chewiness, and resilience, and its growth rate was also the highest compared with the other three kinds of MF dry samples. NGM-MF had the lowest hardness, cohesion, gumminess, chewiness, and resilience.

### 3.5. Analysis of Browning Characteristics among Different MF Varieties

Color appearance serves as an important basis for categorizing the quality grade of MF in the circulation market and is paramount in judging the quality of MF post-processing. Ideally, MF should exhibit a black or brownish-black hue after processing [47]. *BI* is usually used to evaluate the browning degree of materials [48]. Table 3 shows the color appearance changes of different MF varieties. The *BI* of the NGM-MF sample was the highest, and its color appearance was the blackest, which was most in accordance with market requirements and those of the Chinese Pharmacopoeia. Conversely, the *BI* of the ZSM-MF sample was the lowest, and its color appearance was the least compliant with the requirements of the Chinese Pharmacopoeia. MF browning primarily stems from the Maillard reaction, wherein RS and AC (referring to Table 1 for specifics) serve as the principal substrates. Consequently, 5-HMF is the primary product of the Maillard reaction [12]. Table 3 further illustrates the 5-HMF content in different MF samples, with the NGM-MF sample exhibiting the highest levels, followed by QZM-MF, while the YSM-MF sample had the lowest content of 5-HMF. In addition, the correlation coefficients of hardness, gumminess, chewiness, pH, TSS, TA, SS, RS, SP, AC, CA, PT, 5-HMF, and *BI* were expressed by the Pearson correlation coefficient (r). As shown in Figure 6, hardness, gumminess, chewiness, and pH were positively correlated with *BI* (*p* < 0.01), indicating that the higher values of hardness, gumminess, chewiness, and pH value were associated with lower MF browning degrees. Conversely, TA, SS, and 5-HMF displayed negative correlations with *BI* (*p* < 0.01), implying that elevated TA, SS, and 5-HMF contents corresponded with deeper MF browning. The *BI* of MF showed weak correlations with RS and AC in PM, possibly attributed to the sluggish onset of the Maillard reaction in the early stage of processing; the hydrolysis of SS into RS during processing led to an increase in RS content [12]. Some studies also suggested that minimal sugar consumption can yield significant browning [49]. In addition, the higher content of 5-HMF in MF samples with a greater browning degree indicated an intensified Maillard reaction in these samples. However, the weak correlation between AC content and browning degree may be due to the incomplete AC consumption in MF samples.

### 3.6. Analysis of Citric Acid and Extract Content among Different MF Varieties

Citric acid in MF has strong antioxidant, anti-inflammatory, antibacterial, anti-thrombotic, and other effects. The 2020 Chinese Pharmacopoeia takes CA content as the only quality control index of MF [1,5]. The CA content in dry samples of different varieties of MF is shown in Figure 7. The results revealed significant differences in CA content among the four varieties after processing (*p* < 0.05). The CA content of the four kinds of dried MF samples ranged from 18.18 ± 0.89% to 24.88 ± 0.83%, all of which met the requirement of no less than 12% stipulated in the 2020 Chinese Pharmacopoeia. The CA content of the YSM-MF sample was the highest, followed by the NGM-MF and QZM-MF samples, which were 23.28% and 21.90%, respectively. The ZSM-MF sample had the lowest content of CA, which was mainly caused by the variety of PM, soil, and climate. Chen et al. conducted a comparative analysis on the CA content in MF from different places and batches, and also found that CA content varied with different places and batches [19].

As a medicinal and food homologous product, MF has complex medicinal components. Except citric acid, which is the only definite quality index to measure the quality of MF, many of its components remain unclear. The content of water-soluble extracts is usually used as the index to monitor its quality [1]. The water-soluble extracts in MF mainly include organic acids, sugars, phenolic compounds, trace elements, etc. [5]. The extract content of dry samples from different varieties of MF is shown in Figure 7. The results showed significant differences (*p* < 0.05) in the extract contents of four kinds of MF. The extract contents of dry samples of the four kinds of MF ranged from 63.48 ± 1.09% to 77.88 ± 2.30%. The extract content of ZSM-MF was the lowest, while the extract contents of QZM -MF, YSM-MF, and NGM-MF exceeded 70%, reaching 70.92%, 74.08%, and 77.88%, respectively.

### 3.7. Analysis of TP, TF, and TAC among Different Varieties of PM and MF

MF contains abundant phenolic and flavonoid compounds, which have strong pharmacological activities. Phenolic compounds in MF have antioxidant properties, which can help eliminate free radicals in the body and reduce oxidative damage. They also have antibacterial and anti-inflammatory effects, helping to prevent infection and alleviate inflammation [50]. As shown in Figure 8a, there was a significant difference (*p* < 0.05) in the total phenolic content of the four different PM varieties and their processed MF. In terms of PM, the ZSM samples had the highest TP content, reaching 84.62 ± 6.77 mg GAE/g DM, while the YSM samples had a TP content of 66.78 ± 3.27 mg GAE/g DM. Then, for the QZM samples, the TP content was 33.79 ± 1.90 mg GAE/g DM, while the NGM samples had the lowest TP content, which was 23.66 ± 0.69 mg GAE/g DM. The TP content in the four different MF samples ranked from high to low as follows: ZSM-MF (59.65 ± 5.91 mg GAE/g DM), YSM-MF (27.48 ± 2.90 mg GAE/g DM), QZM-MF (13.82 ± 1.07 mg GAE/g DM), and NGM-MF (13.79 ± 1.65 mg GAE/g DM). These results are consistent with their corresponding PM varieties. However, compared with the TP content in PM, the TP content in MF was lower. This may be due to the thermal degradation of TP in PM caused by high-temperature conditions during the processing, resulting in a decrease in TP content in MF [51].

Flavonoids have high medicinal value due to their broad-spectrum pharmacological activity and low toxicity. They can eliminate and reduce the production of free radicals, and have effects such as lowering blood sugar, anti-tumor, and anti-lipid peroxidation [5]. As shown in Figure 8b, there were significant differences (*p* < 0.05) in the TF content of the four different PM varieties and their processed MF. The TF content in the ZSM samples was 62.9 ± 2.26 mg RUE/g DM, while the TF content in the YSM and NGM samples was 48.7 ± 3.64 mg RUE/g DM and 10.3 ± 0.68 mg RUE/g DM, respectively, and the TF content in the QZM samples was the lowest at 6.82 ± 1.05 mg RUE/g DM. The ranking of TF content in the four different MF samples was consistent with their corresponding PM varieties, with ZSM-MF (42.18 ± 4.35 mg RUE/g DM), YSM-MF (17.93 ± 2.57 mg RUE/g DM), NGM-MF (6.91 ± 0.54 mg RUE/g DM), and QZM-MF (4.60 ± 0.55 mg RUE/g DM) ranking from high to low, respectively. The TF content in the four different samples of MF was lower than that in their corresponding PM. This may be due to the thermal degradation of flavonoids in PM caused by high-temperature conditions during processing, resulting in a decrease in TF content in MF [52].

Figure 8c illustrates the TAC of different PM varieties and processed MF. The results indicate significant differences (*p* < 0.05) in the TAC between the four PM varieties and their processed MF. The overall antioxidant capacity of the PM sample was stronger than that of the MF sample, and the PM sample contained more phenolic and flavonoid compounds. Therefore, it can be seen that the changes in the TAC of PM and MF were closely related to their TP and TF content [53]. Among the PM samples, the ZSM samples had the strongest TAC at 231.95 ± 4.75 mg TE/g DM; next was the YSM samples, with a TAC of 107.68 ± 2.53 mg TE/g DM. The TAC of the QZM samples and NGM samples showed slight variations, with values of 75.79 ± 1.96 mg TE/g DM and 67.85 ± 1.62 mg TE/g DM, respectively. The TAC of MF samples was consistent with that of PM samples, with ZSM-MF (123.79 ± 5.15 mg TE/g DM), YSM-MF (89.97 ± 3.11 mg TE/g DM), QZM-MF (62.15 ± 1.58 mg TE/g DM), and NGM-MF (58.24 ± 2.60 mg TE/g DM) in descending order. It can be seen that the variety had a significant impact on the TAC of both PM and MF, and the TAC of MF varied among different varieties.

## 4. Conclusions

This study showed significant differences in physicochemical properties among PM varieties from different regions (*p* < 0.05), which directly affected the efficiency and quality of processing MF. ZSM exhibited a hard texture that made it difficult for water to migrate, prolonging the processing time. However, the total phenolic and flavonoid content of fresh ZSM were nearly 2–5 times higher than that of the other three types of PM, along with an antioxidant capacity that was 2–3 times greater. ZSM-MF also retained the highest levels of total phenolics and total flavonoids and the highest antioxidant capacity, making it ideal for extracting antioxidant components for enriching food production. YSM, with antioxidant capacity trailing only ZSM, boasted the shortest processing time. Additionally, YSM-MF exhibited the highest citric acid content, rendering it suitable for edible MF development. QZM, distinguished by its high titratable acid content, lowest pH value, and pronounced acidity, can serve as a raw material for producing PM juice and wine. NGM stood out for its elevated soluble solid and soluble sugar contents, along with a relatively shorter processing time. Notably, NGM-MF showcased exceptional performance in appearance, color, rehydration ability, and citric acid content, aligning with the requirements of the 2020 Chinese Pharmacopoeia and making it suitable for traditional Chinese medicine MF production. This study provides a theoretical basis for the selection of PM raw materials for different uses of MF, and also provides a reference for the comprehensive utilization of PM. At the same time, this study found that the quality of PM may also be affected by soil, temperature, precipitation, and other conditions in different production areas. Subsequent research should comprehensively consider the impact of these conditions.

## Figures and Tables

**Figure 1 foods-13-01377-f001:**
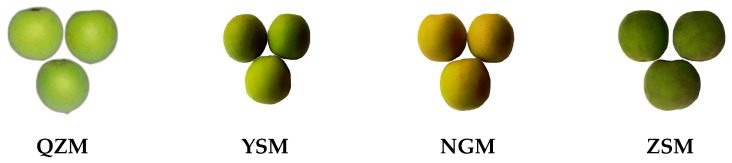
Different fresh PM varieties.

**Figure 2 foods-13-01377-f002:**
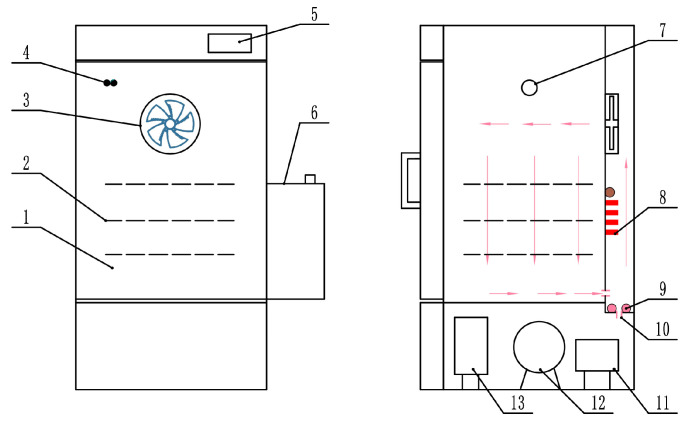
Structure diagram of hot-air drying equipment based on precise temperature and humidity control: 1: drying chamber; 2: material rack; 3: circulating fan; 4: temperature and humidity sensor; 5: touch screen; 6: water tank; 7: test hole; 8: finned heater; 9: steam heating pipe; 10: water inlet; 11: liquid reservoir; 12: compressor; 13: condenser.

**Figure 3 foods-13-01377-f003:**
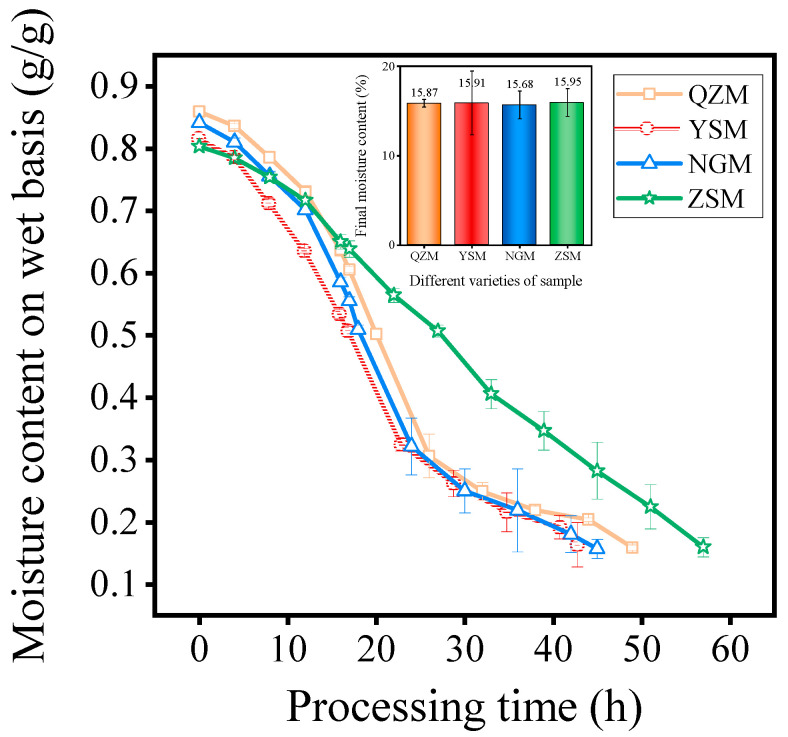
Curves of moisture content on wet basis and processing time of different PM varieties.

**Figure 4 foods-13-01377-f004:**
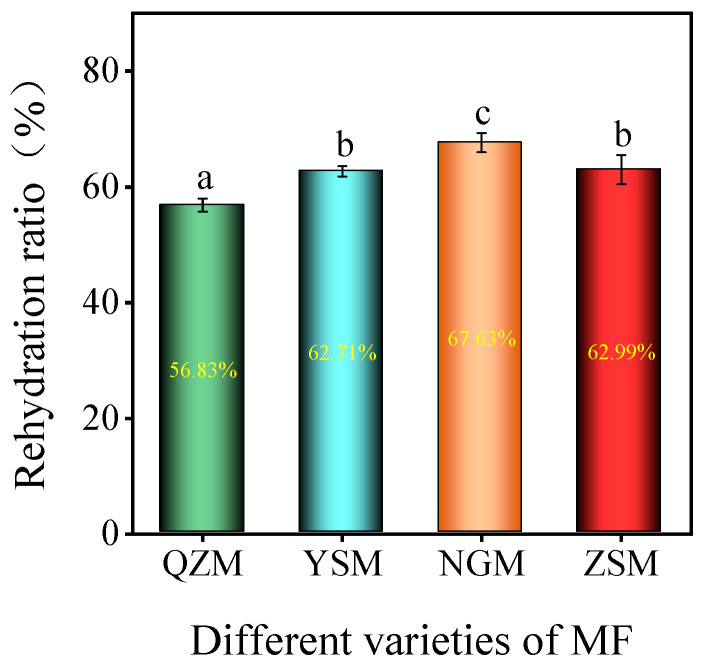
The RR of different varieties of MF (the different lowercase letters reveal significant differences (*p* < 0.05) according to the Duncan test).

**Figure 5 foods-13-01377-f005:**
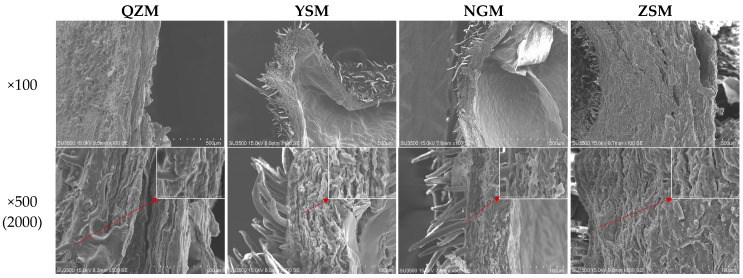
The microstructure of MF processed from different varieties at 100 and 500 (2000) magnification.

**Figure 6 foods-13-01377-f006:**
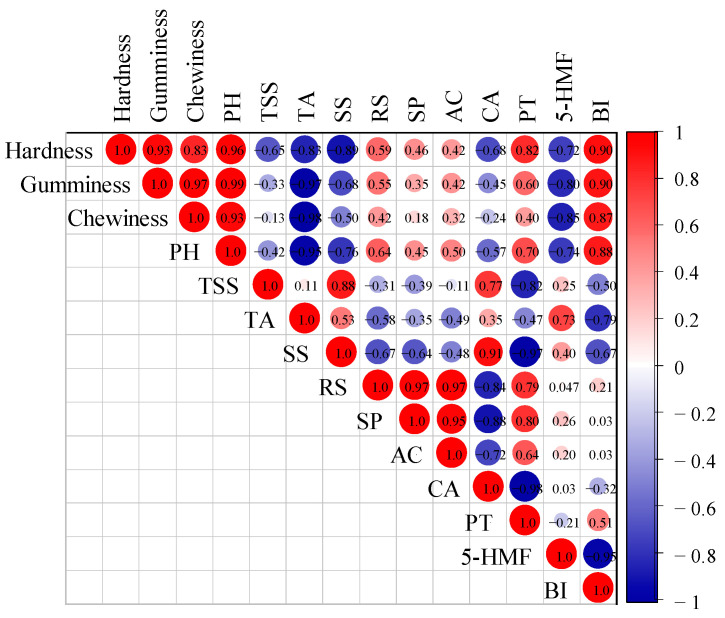
Correlation matrix between the determined parameters.

**Figure 7 foods-13-01377-f007:**
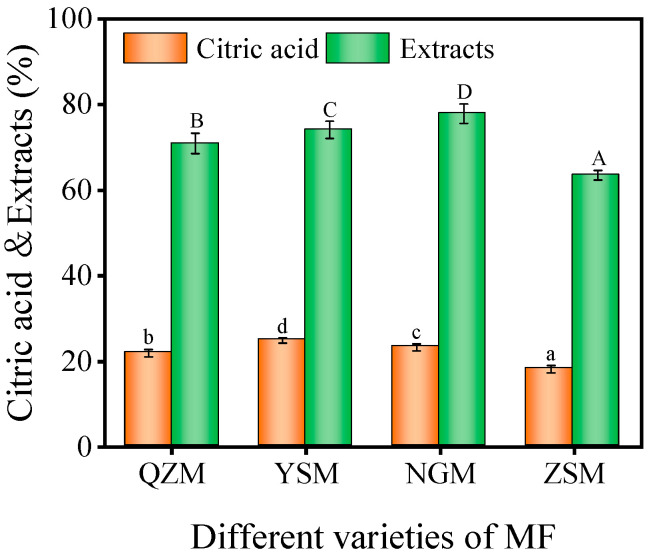
CA and extract content of different varieties of MF (the different lowercase letters reveal significant differences (*p* < 0.05) of different MF in CA content, while the uppercase reveals significant differences (*p* < 0.05) of different MF in extract content).

**Figure 8 foods-13-01377-f008:**
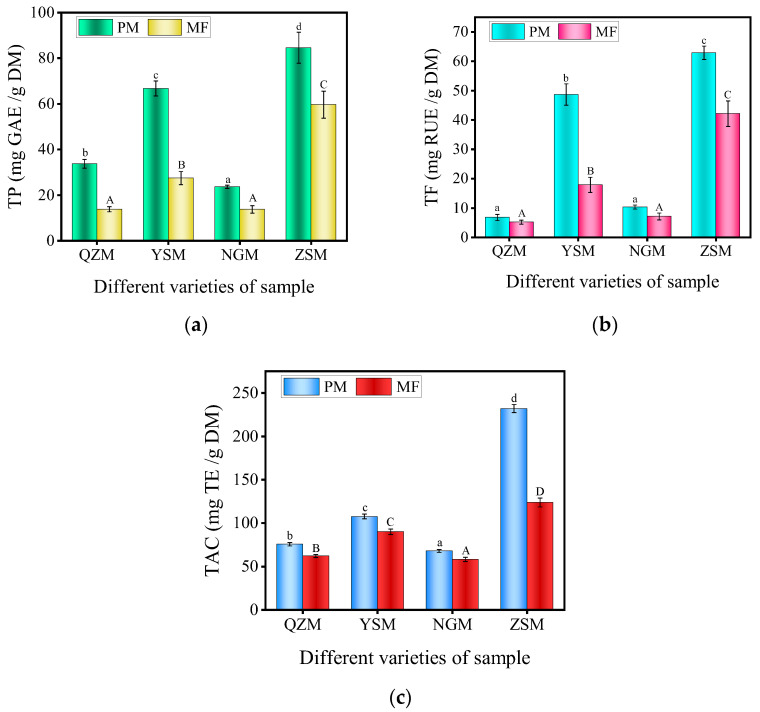
(**a**) TP content of different varieties of PM and MF; (**b**) TF content of different varieties of PM and MF; (**c**) TAC of different varieties of PM and MF. (The different lowercase letters in same picture reveal significant differences (*p* < 0.05) of PM, while the uppercase reveals significant differences (*p* < 0.05) of MF).

**Table 1 foods-13-01377-t001:** Physicochemical indicators of different PM varieties.

PM Variety	MC (%)	pH	TSS (%)	TA (g/100g)	SS (mg/g)	RS (mg/g)	SP (mg/g)	AC (u mol/g)	CA (mg/g)
QZM	85.89 ± 0.91 ^c^	2.19 ± 0.01 ^a^	8.13 ± 0.12 ^b^	6.61 ± 0.20 ^d^	26.28 ± 1.34 ^b^	3.29 ± 0.07 ^b^	2.85 ± 0.07 ^b^	3.37 ± 0.10 ^b^	6.08 ± 0.04 ^b^
YSM	81.48 ± 0.68 ^a^	2.37 ± 0.02 ^c^	10.00 ± 0.10 ^c^	5.53 ± 0.11 ^b^	31.33 ± 0.29 ^c^	2.79 ± 0.06 ^a^	2.46 ± 0.15 ^a^	2.32 ± 0.12 ^a^	6.18 ± 0.02 ^d^
NGM	84.11 ± 0.55 ^b^	2.26 ± 0.01 ^b^	10.90 ± 0.10 ^d^	5.86 ± 0.17 ^c^	33.09 ± 0.45 ^d^	4.07 ± 0.09 ^c^	3.03 ± 0.12 ^b^	5.89 ± 0.08 ^c^	6.12 ± 0.03 ^c^
ZSM	80.34 ± 0.86 ^a^	2.63 ± 0.02 ^d^	7.80 ± 0.10 ^a^	4.84 ± 0.12 ^a^	16.31 ± 0.07 ^a^	4.84 ± 0.06 ^d^	3.26 ± 0.12 ^d^	6.54 ± 0.11 ^d^	6.00 ± 0.02 ^a^

Note: the different letters in the same column reveal significant differences (*p* < 0.05) according to the Duncan test.

**Table 2 foods-13-01377-t002:** Texture characteristic parameters of different varieties of PM and MF.

Type	Variety	Hardness (N)	Springiness (%)	Cohesiveness (%)	Gumminess (N)	Chewiness (mJ)	Resilience (%)
PM	QZM	40.04 ± 1.66 ^a^	52.02 ± 1.76 ^a^	35.34 ± 2.29 ^a^	14.18 ± 1.48 ^a^	7.39 ± 0.96 ^a^	15.41 ± 1.38 ^a^
	YSM	44.60 ± 1.61 ^b^	61.15 ± 1.20 ^b^	47.30 ± 1.31 ^b^	21.10 ± 0.88 ^c^	12.91 ± 0.74 ^c^	19.52 ± 1.50 ^bc^
	NGM	37.04 ± 1.75 ^a^	58.83 ± 1.43 ^b^	45.46 ± 0.93 ^b^	16.84 ± 1.00 ^b^	9.91 ± 0.71 ^b^	18.94 ± 0.65 ^b^
	ZSM	59.67 ± 5.89 ^c^	54.73 ± 4.09 ^a^	45.74 ± 2.09 ^b^	27.20 ± 1.64 ^d^	14.87 ± 1.11 ^d^	20.80 ± 0.52 ^c^
	Average	44.75	56.16	42.56	19.20	10.84	18.30
MF	QZM-MF	88.52 ± 7.23 ^B^	57.09 ± 3.08 ^A^	65.54 ± 4.04 ^B^	58.12 ± 7.36 ^B^	33.24 ± 5.25 ^A^	33.39 ± 2.26 ^B^
	YSM-MF	107.35 ± 6.24 ^C^	60.57 ± 5.00 ^A^	67.81 ± 2.77 ^B^	72.85 ± 6.00 ^C^	44.32 ± 6.82 ^B^	40.02 ± 1.18 ^C^
	NGM-MF	77.84 ± 2.85 ^A^	57.89 ± 3.03 ^A^	60.94 ± 2.42 ^A^	47.49 ± 3.50 ^A^	27.55 ± 3.13 ^A^	29.55 ± 1.25 ^A^
	ZSM-MF	247.58 ± 9.44 ^D^	85.95 ± 2.09 ^B^	90.06 ± 1.24 ^C^	223.03 ± 11.02 ^D^	191.71 ± 11.16 ^C^	79.45 ± 1.69 ^D^
	Average	124.49	64.54	69.96	94.50	69.2	43.82

Note: the different lowercase letters in the same column reveal significant differences (*p* < 0.05) of PM, while the different uppercase in the same column reveals significant differences (*p* < 0.05) of MF.

**Table 3 foods-13-01377-t003:** Color parameters and 5-HMF content of different varieties of MF.

Variety	Image	*L**	*a**	*b**	Δ*E*	*BI*	5-HMF
QZM	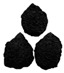	23.34 ± 1.37 ^a^	1.73 ± 0.24 ^a^	4.46 ± 0.26 ^a^	55.31 ± 1.02 ^c^	25.96 ± 1.60 ^b^	1.69 ± 0.02 ^c^
YSM	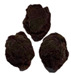	26.61 ± 1.80 ^bc^	2.63 ± 0.59 ^b^	6.45 ± 0.32 ^b^	51.87 ± 1.24 ^b^	34.70 ± 2.98 ^c^	1.15 ± 0.01 ^a^
NGM	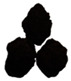	27.67 ± 1.42 ^c^	1.89 ± 0.29 ^a^	4.53 ± 0.34 ^a^	52.50 ± 0.98 ^b^	22.61 ± 2.06 ^a^	1.77 ± 0.01 ^d^
ZSM	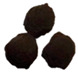	25.51 ± 1.78 ^b^	3.30 ± 0.46 ^c^	6.47 ± 0.71 ^b^	44.26 ± 1.31 ^a^	38.46 ± 4.48 ^d^	1.23 ± 0.01 ^b^

Note: the different letters in the same column reveal significant differences (*p* < 0.05) according to the Duncan test.

## Data Availability

The data presented in this study are available on request from the corresponding author. The data are not publicly available due to privacy.

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
