# Peer review of "Effects of Different Varieties on Physicochemical Properties, Browning Characteristics, and Quality Attributes of Mume fructus (Wumei)"

_foods, 2024, doi:10.3390/foods13091377_

Round 1

Reviewer 1 Report

Comments and Suggestions for Authors

Present study compared the Effects of Different Varieties on Physicochemical Properties, Browning Characteristics and Quality Attributes of Mume Fructus (Wumei).

This paper presents extensive research. There are spelling errors in many sentences in the MS. Especially, figures and tables are explained n a limited manner. In the method section, some methods are given incompletely or incorrectly. The discussion should be improved by explaining and interpreting the results chronologically. My advice to author(s) is to reconsider the entire manuscript (MS). The revision suggestions are listed below:

Line 15: The name of the varieties used is provided in parentheses.

Lines 118-125: Why was the moisture content determined after the samples were washed? Isn't it more accurate to determine it before washing? Because there is a high probability of an increase in moisture content after washing.

Line 173: In PMs, was the color measured in one direction in each fruit? Generally, measurements are taken in 2 or 3 different directions on fruits and their average represents 1 repetition. Color-related deficiencies should be explained in detail.

Lines 256, 266 and 277: Information about the device used in determining TP, TF and TAC and its brand, model etc. should be given.

Line 286: Statistical analysis section should be explained in detail. How was parametric testing done without testing whether the data showed a normal distribution? The method should be written clearly, emphasizing the dependent and independent variables. If there are significant differences when ANOVA analysis is performed, Tukey HSD multiple comparison test should be performed, and grouping should be done. These deficiencies should be clarified by the authors.

For example, correlation analysis was performed, but it was not mentioned in the statistical analysis section.

Lines 294-303: The paragraph given here is unnecessary. Because it is general information. Therefore it should be removed from here. If it is to be given, it should be given in the introduction section.

Lines 305-317: Changes occurring in PMs are given. However, no information is given about the factors affecting these changes. Therefore, the discussion about the changes occurring in PMs is very weak. Authors should explain the reasons for the changes with references.

The explanation given above applies in sections 3.2 and 3.3.

Line 376: The authors' presentation style is incorrect. In MS, you cannot start writing directly by giving a table. The text should be written first, the relevant Table should be cited, and then the table(s) should be given.

Line 376: According to Table 2, increases were generally observed in the parameters examined after processing. However, the authors did not explain the reasons for these increases in the text. Therefore, the discussion about the changes occurring in MFs is very weak.

Lines 398 and 400: The authors' presentation style is incorrect. In MS, you cannot start writing directly by giving a table and figure. The text should be written first, the relevant Table should be cited, and then the table(s) should be given.

Comments on the Quality of English Language

There are grammatical and spelling errors in some parts of MS. Please reconsider.

Reviewer 2 Report

Comments and Suggestions for Authors

In the presented manuscript, the authors have studied the effects of variety on the physicochemical properties and quality of Mume Fructus. The article is generally well-written and well-structured. However, the research conducted by the authors should be considered preliminary and of limited applicability.

1.       The authors compared fruits only from the same harvest year, which undermines the practical utility of the obtained results.

2.       In the section regarding data analysis, it should be specified which test was used to determine the significance of differences between means.

3.       Abbreviations of variables should be explained below tables and figures.

4.       What was the moisture content of the fruits used for texture and color analysis?

5.       Description of Fig 2. Please be consequent and use small or capital letters.

6.       Fig. 3: On the Y-axis, is it moisture content or moisture ratio? Where are the results of the moisture ratio (MR)?

7.       The unnecessary bracket in the first row of Table 2 should be removed.

8.       Tables and figures in the text should appear after they are cited, not before.

9.       What was the size (diameter) of the fruit?

10.   Please include a picture of the fruit after drying. The conclusions should necessarily include limitations arising from the conducted research.

11. In the keywords the "drying" word should be used. 

Reviewer 3 Report

Comments and Suggestions for Authors

1. The manuscript has to be fully rewritten, regarding the scientific style of writing. 

2. The Abstract and Introduction are way too long. So many unnecessary details, written in a patchwork style. 

3. p1, l39: "The 2020 Chinese Pharmacopoeia establishes citric acid content as the quality control standard for MF." Add reference. 

4. The reference 5 has nothing to do with health benefits of MF. 

5. p2, l55: "It is generally believed..." Not really a scientific construction of a sentence. 

6. p2, l57: NEB, which ones?

7. p2, l75-6: "Hence, the quality of processed MF can be influenced by the different varieties of PM." Well-known conclusion. Introduction is not a chapter for conclusions.

8. p2, l77-86: Various production regions: Are we talking about the same or various cultivar(s)? The question is: are we talking about one or more variables, which influence the chemical composition?

9.  p3, l107-8: "Fresh PM of various varieties, including Qingzhu Mei (QZM), Yesheng Mei (YSM), Nangao Mei (NGM), and Zhaoshui Mei (ZSM), was shown in Figure 1." Improper subchapter's initiual sentence. 

10. p3, l108-11: GPS coordinates are missing. It is very important to add GPS coordinates, since these 4 cultivars were grown at various locations. Thus it is of extreme importance to underline that, since all differences in chemical composition, dye, texture, etc. stem from various cultivars, but also from various locations, i.e. different soil composition and climate conditions. 

11. p3, l123-5: "The moisture con- 123 tent of QZM, YSM, NGM, and ZSM was measured to be 85.89 ± 0.8%, 81.48 ± 0.71%, 84.11 124 ± 0.77%, and 80.34 ± 0.65%, respectively".

12. p3 , l115: How long was the period between harvest and processing?

13. p3, l135-6: monolayer?

14. p4, Figure 2: Provide better and clearer scheme. 

15. p4, l156: Extracted how?

16: p4, subchapter 2.4: Clumsy written. 

17. Place equation (4) above in the text.

18. p6, l216: How was the powder obtained?

19. p6, l226: How were the identification and quantification performed?

20. p7, l258: How was the MF extract performed? 

21. p7, l274: "The TF content was expressed as milligrams of gallic acid equivalent per gram of dry matter (mg GAE/g DM), based on the equivalent of rutin." I do not understand that: gallic acid is not a flavonoid. 

22: p7, l297-304: I find uncommon to make here analogy with Magnolia officinalis, Polygonatum and Gastrodia elata.

23. p8, l319: "they are also closely linked to the growing environment of PM". This is correct, but since it's not a objective of this work, and since 4 cultivars were grown at 3 different location, this sentence should be erased.

24. p8, Figure 3: Figure inset is unreadable and unnecessary.

25. Subchapter cannot begin with a Figure or Table: Figure 3, Figure 4, Figure 5, Table 2, Table 3, Figure 6. 

26. Data in Figure 4 may be presented differently.

27. p9, l353-5: "The RR of MF samples processed from different PM varieties with THPC-HAD was shown in Figure 4. The results revealed that the RR of the four dried MF samples ranged from 56.83 ± 1.15% to 67.63 ± 1.66%." These values are not presented in Figure 4. 

28. p10, l361-4: "As shown in Figure 5, SEM analysis revealed that QZM-MF samples had a tightly packed fruit skin, thicker middle fruit cortex, tight and smooth interlaminar connections, and minimal pore grooves, which hindered water infiltration and resulted in the lowest rehydration ability [39]." I do not see this on SEM micrographs. 

29. p10, l388: "The hardness, elasticity, cohesion, gumminess, chewiness and resilience of processed MF samples were significantly higher than those of fresh MF samples." It does not stand for springiness/elasticity. 

30. p14, Figure 8: All flavonoids are phenolic compounds, so it does not make sense that amount of flavonoids is higher than amount of phenolics. 

Round 2

Reviewer 1 Report

Comments and Suggestions for Authors

The corrections I suggested in the 1st revision were made by the authors. Good luck.

Author Response

Thank you very much.

Reviewer 2 Report

Comments and Suggestions for Authors

The authros correctted the manuscript accordingly.

Author Response

Thank you very much.

Reviewer 3 Report

Comments and Suggestions for Authors

The manuscript has been improved. Nevertheless, you need to add into the manuscript the details you have explained in the Report notes. For instance: my comment 18 ("How was the powder obtained?"), you have explained nicely in the Report notes, but you havn't added any details into the manuscript. Please do so, regarding all previous comments.

I am not satisfied with Response 30. You have written "And Figure 8 mainly compared the content of related compounds between PM and processed MF." That is correct. But if you compare the Figure 8a and Figure 8b, you can clearly see that amount of total flavonoids of, for instance, PM-YSM is higher compared to the amount of total phenolics (in PM-YSM), and that doesn't make sense.  I suggest that you re-perform the calibrations, and recalculate these parameters.  And why the unit of total flavonoid contents is mg CAE/g DM? It is either catechin equivalent or should be mg RUE/g DM?
